# Spinal Cord Injury and Postdural Puncture Headache following Cervical Interlaminar Epidural Steroid Injection: A Case Report

**DOI:** 10.3390/medicina58091237

**Published:** 2022-09-07

**Authors:** Hyung Joon Park, Heezoo Kim, Sung Jin Jeong, Jae Hak Lee, Sang Sik Choi, Chung Hun Lee

**Affiliations:** 1Department of Anesthesiology and Pain Medicine, Hanyang University Guri Hospital, Gyeongchun Road 153, Guri 11923, Korea; 2Department of Anesthesiology and Pain Medicine, Korea University Medical Center, Guro Hospital, Gurodong Road 148, Guro-gu, Seoul 08308, Korea

**Keywords:** spinal cord injury, postdural puncture headache, cervical epidural steroid injection

## Abstract

*Background*: Cervical interlaminar epidural steroid injection (CIESI) is increasingly used as an interventional treatment for pain originating from the cervical spine. However, serious neurological complications may occur during CIESI because of direct nerve damage following inappropriate needle placement. *Case report*: A 35-year-old woman presented with posterior neck pain radiating to the left upper arm. Cervical magnetic resonance imaging (MRI) revealed left C6 nerve impingement. CIESI under fluoroscopic guidance was performed at another hospital using the left C5/6 interlaminar approach. Immediately after the procedure, the patient experienced dizziness, decreased blood pressure, motor weakness in the left upper arm, and sensory loss. She visited our emergency department with postdural puncture headache (PDPH) that worsened after the procedure. Post-admission cervical MRI revealed intramedullary T2 high signal intensity and cord swelling from the C4/5 to C6/7 levels; thus, a diagnosis of spinal cord injury was made. The patient’s PDPH spontaneously improved after 48 h. However, despite conservative treatment with steroids, the decrease in abduction of the left fifth finger and loss of sensation in the dorsum of the left hand persisted for up to 6 months after the procedure. As noticed in the follow-up MRI performed 6 months post-procedure, the T2 high signal intensity in the left intramedullary region had decreased compared to that observed previously; however, cord swelling persisted. Furthermore, left C7/8 radiculopathy with acute denervation was confirmed by electromyography performed 6 months after the procedure. *Conclusions*: Fluoroscopy does not guarantee the prevention of spinal cord penetration during CIESI. Moreover, persistent neurological deficits may occur, particularly due to intrathecal perforation or drug administration during CIESI. Therefore, in accordance with the recommendations of the Multisociety Pain Workgroup, we recommend performing CIESI at the C6/7 or C7/T1 levels, where the epidural space is relatively large, rather than at the C5/6 level or higher.

## 1. Introduction

Cervical epidural steroid injection (CESI) is increasingly used as a medical interventional treatment for pain originating from the cervical spine, particularly for cervical radicular pain [1]. As the number of patients complaining of cervical radicular pain increases with age, the frequency of CESI treatment also increases [1].

Cervical interlaminar epidural steroid injection (CIESI) is used more often than cervical transforaminal epidural steroid injection (CTESI) because of the lower risk of intravascular injection and lower complication rates [2,3]. A literature review from 1990 to 2010 reported cases of permanent spinal cord injury in six and fifteen patients after CIESI and CTESI, respectively [3,4,5]. However, although the frequency is relatively low, direct nerve damage may occur during CIESI because of inappropriate needle placement and serious neurological complications may result from space-occupying lesions, such as hematomas or abscesses [6]. Depending on the height of the involved cervical spine as well as the type and volume of infusion used, direct spinal cord injury after CIESI can cause various symptoms. This may lead to irreversible consequences, such as hemiplegia and death in severe cases [7,8].

Several studies on measures that can be used to reduce the risk of nerve damage, hematomas, and infections during CIESI have been conducted [9,10,11]. In addition, the Multisociety Pain Workgroup in 2015 and Schneider et al. in 2018 proposed recommendations to reduce the risk of complications, which include using the contralateral oblique technique, limiting the use of sedation, performing the injection at C6/7 or below, and using sterile techniques [6,12]. However, in practice, these recommendations are not compulsory; thus, they are not always followed. Therefore, it is important to understand the symptoms and potential risks of epidural injections, and to improve awareness regarding potential complications.

This report describes a case of spinal cord injury that occurred after CIESI was performed without following the above-described recommendations.

## 2. Case Report

This study was approved by the Institutional Review Board of the Korea University Medical Center, Guro Hospital, Seoul, Republic of Korea (2022GR0107), on 16 February 2022.

Before visiting our hospital, a 35-year-old woman with no specific medical history visited a spine hospital complaining of posterior neck pain radiating to the left upper arm. These symptoms appeared 6 months before she was admitted at the spinal hospital and had worsened to a Visual Analog Scale (VAS) score of 6 at the time of admission. No motor weakness or sensory changes were observed upon physical examination at the time of admission. However, numbness was observed in the left arm during the Spurling test. Magnetic resonance imaging (MRI) of the cervical spine showed a herniated intervertebral disc at the C5/6 level and left C6 nerve impingement due to left foraminal extrusion (Figure 1).

Subsequently, for left C6 impingement, the patient underwent CIESI under fluoroscopy using the C5/6 interlaminar approach. According to the medical records, when the Tuohy needle was inserted epidurally during the procedure, the patient experienced severe pain, described as burning and electric, in the left arm. Subsequently, a mixture of 3 mL of 1% lidocaine and 1 mL of dexamethasone was injected in the epidural space. Immediately after the procedure, the patient was in an alert mental state; however, severe dizziness, reduced blood pressure (BP 60/40), reduction of left arm motor function to motor grade 0, and loss of sensory function other than touch were observed. Blood pressure instability was relieved within approximately 1 h after the procedure with the administration of vasopressors, inotropes, and anticholinergics. However, motor weakness and sensory loss persisted in her left arm. Furthermore, 6 h after the procedure, the patient developed postdural puncture headache (PDPH) with a VAS score of approximately 7, after which she presented to the emergency department of our hospital. At the time of her visit, her vital signs were normal. On the second day of CIESI, the PDPH-related VAS score was 7. Upon physical examination, the manually tested muscle power of the left upper extremity was 2/5 proximally and 3/5 distally. Pressure and temperature discrimination, light touch, and vibration sensations decreased in the left upper arm. The remaining extremities and facial areas had intact motor and sensory functions. The patient complained of severe PDPH; therefore, the presence of gait disturbances could not be confirmed. Contrast-enhanced brain and cervical spine MRIs were performed on the second day of the procedure. After brain MRI, the assessment by the radiologist was of a “probable small amount of parafalcine subdural hematoma (SDH), left” (Figure 2).

After consulting with neurosurgeons, no clear findings or symptoms of SDH other than PDPH were found; therefore, conservative treatment was provided. When the cervical spine MRI scans obtained on the second day of the procedure were compared with those obtained immediately before the procedure, intramedullary T2 high signal intensity and cord swelling from the C4/5 to C6/7 levels were observed, and the patient was diagnosed with a spinal cord injury (Figure 3).

After confirming the findings of the cervical spine MRI, drug treatment was initiated with steroid pulse therapy (intravenous methylprednisolone 1 g daily), pregabalin 75 mg twice daily, duloxetine 30 mg once daily, and tramadol/acetaminophen 75/650 mg twice daily. As the symptoms of PDPH nearly resolved at 48 h after the procedure, an epidural blood patch was not performed. The above-mentioned drugs, including steroids, were continued for another two weeks. After 2 weeks, the manually tested left upper extremity muscle power recovered slightly to 3/5 proximally and 4/5 distally, with a slight improvement in sensory change; gait disturbances were not observed. Therefore, the patient opted to be followed up on an outpatient basis.

Medications, excluding steroids, were continued for 1 month after the procedure. The patient reported that their left upper arm motor weakness improved slightly (4/5); however, sensory changes persisted. Two months after the procedure, shoulder strength was almost fully restored (5/5), left-hand motor weakness slightly improved further (4/5), and sensory function had almost returned to normal; however, the patient complained of persistent paresthesia in her left hand. At 3 months after the procedure, her left-hand motor weakness showed almost complete recovery (5/5); however, abduction of her fifth finger diminished to 4/5, and the decreased sensation in the dorsum of the left hand persisted. Six months after the procedure, abduction of the fifth finger continued to decrease and sensory deterioration in the dorsum of the left hand remained. As detected during the follow-up MRI performed 6 months post-procedure, the T2 high signal intensity in the left central intramedullary region decreased compared to that observed previously; however, spinal cord swelling was still present, albeit reduced (Figure 4).

In addition, left C7/8 radiculopathy with acute denervation was confirmed by electromyography that was performed at 6 months after the procedure.

## 3. Discussion

This report describes a case of spinal cord injury that occurred during CIESI at the C5/6 level performed following left C6 nerve root compression.

CIESI is an effective method for managing cervical radicular pain; however, it can cause several adverse effects. The reported rate of complications due to CIESI varies from 0 to 16.8% [3]. Using a midline or paramedian approach from the rear, it targets the epidural space between the dura mater anteriorly and ligamentum flavum posteriorly. The target of the needle tip was within a few millimeters of the spinal canal. Dural puncture and cord penetration can occur if the needle is advanced too anteriorly.

The injection of certain drugs without awareness of these potential complications can have disastrous consequences. Intrathecal administration of anesthetics to the cervical spine increases the risk of high spinal anesthesia. This may result in respiratory failure that requires ventilation, hemodynamic instability, and cardiac arrest in severe cases. Even if intrathecal drugs are not administered, the needle is placed in the spinal canal during the CIESI procedure, and dural puncture may cause spinal headache and direct spinal cord injury, leading to permanent nerve damage (e.g., hemiplegia).

In 1998, Hodges et al. reported two patients with sequelae from spinal cord injuries due to dural puncture and intrathecal epidural drug administration after C5/6 interlaminar injection under sedation [7]. In both cases, signs of intrinsic spinal cord injury at the injection level were observed when comparing the MRIs obtained immediately after the procedure with the preprocedural MRIs. After the procedure, the first patient experienced severe pain in the C7 distribution in the right arm towards the treatment site, motor weakness in the right C7 innervation, numbness in the front of the right thigh, and paresthesia in the C7/8 distribution. Although C5/6 anterior cervical microdisc resection was performed 1 month after the procedure, sensory abnormalities in the C7/8 distribution persisted for 6 months. The second patient also complained of severe refractory reflex sympathetic dystrophy of the left arm, which was the treatment site, and paresthesia of the right thigh 9 months after the procedure.

In 2014, Maddela et al. reported a case of a patient who developed hemiparesis and facial sensory loss after C5/6 interlaminar injection with sedation [8]. MRI revealed a T2-weighted high signal from C6/7 to the base of the brain immediately after the procedure. The patient reported facial sensation recovery after 3 months; however, motor function in the right leg continued to decrease to grade 4 in the manual muscle test for up to 3 months.

The patient in the current report also received interlaminar epidural steroid injections at the C5/6 level. We assumed that the physician chose this level to access the area closest to the patient’s lesion. This approach was the closest to the area of nerve damage (left C6 nerve root compression). The patient showed cord injury at the C4/5 to C6/7 levels after the procedure, with no facial sensory loss or motor weakness in the lower extremities. However, she showed decreased abduction of the left fifth finger and decreased sensation in the left hand until 6 months after the procedure.

The two previous reports and our current report are similar in that the needle entered the C5/6 interlaminar space during CIESI. This was probably because this area was closest to the patients’ lesions. However, the mid-portion of the cervical spinal cord allows it to be positioned closer to the epidural space because of its natural bulge. Therefore, when CIESI is performed at the C6/7 level or a higher level, the likelihood of spinal cord interruption increases because of the relatively thin epidural layer compared to that of the lower cervical regions [7,8]. There could also be instances in which the loss-of-resistance (LOR) technique for identifying the epidural space during CIESI may lack specificity. This is because the ligamentum flavum, which guides the LOR technique, may be deficient with CIESI. The proportion of ligamentum flavum with a midline fissure was 51% in the C7/T1 epidural space, 65% in the C6/7 epidural space, and 74% in the C5/6 epidural space [13]. The lack of feedback due to these LOR difficulties may also explain why CIESI can increase the likelihood of dural puncture when performed at the C6/C7 level and higher.

Therefore, the Multisociety Pain Workgroup and Schneider et al. recommend performing CIESI at or below the C6/7 level to reduce the risk of spinal cord injury [6,12]. Unless otherwise specified, the C6/7 or C7/T1 interlaminar space is the preferred injection site because of the relatively large epidural space at this level compared to that of the other levels of the cervical spine. In addition, it was reported that even if the injection is performed at C7/T1, the drug can spread to C5/6 in 92.9% of patients if a 5 mL volume of the injection solution is used and in 97.6% of patients if a 10 mL volume of the injection solution is used [14]. Thus, this previous report supported performing CESI at C7/T1, even if the lesion is located at C5/6.

However, objections to this approach do exist. Schultz et al. reported no difference in complications according to spinal level (C4/5, C5/6, C6/7, and C7/T1) in 12,168 CIESI procedures performed at a single institution [15]. In addition, Manchikanti et al. reported no difference in complications according to the level (C5/6, C6/7, or C7/T1) when they retrospectively analyzed 4396 patients who underwent CIESI [16].

On the other hand, these studies were retrospective and involved highly skilled doctors from a single institution. Therefore, it is unknown whether there is a difference in complications among doctors with lower skill levels. In fact, in two previous case reports [7,8], in which spinal cord injury occurred as a fatal complication, an experienced anesthesiologist performed the procedure using an interlaminar approach at C5/6. In a study conducted by Manchikanti et al., the dural puncture rates were 1.8% for C5/6, 0.87% for C6/7, and 1.71% for C7/T1 [16]. However, the diameter of the posterior epidural space is approximately 1.5–2 mm at the C7 level and decreases at higher levels [17]. Moreover, in a study conducted by Manchikanti et al., the rate of dural puncture was 0.9% higher when CESI was performed at the C7/T1 level than at the C6/7 level, and the dural puncture rate was similar between the C7/T1 and C5/6 levels [16]. One of the reasons for this finding, as mentioned by Manchikanti et al., is that CIESI was performed from a lower position away from the surgical site in patients who underwent cervical spine surgery; however, no exclusion from the study analysis was mentioned [16]. In addition, Schultz et al. and Manchikanti et al. did not mention the proportion of patients who underwent cervical spine surgery at the C5/6, C6/7, and C7/T1 levels and did not correct the results. In our opinion, the rate of dural puncture may increase when CIESI is performed in patients who have undergone surgery for postoperative adhesion. Therefore, unlike the previously stated anatomical common sense in the study by Manchikanti et al., the reason the epidural puncture rate at C6/7 was 0.9% lower than that at C7/T1 may be because patients with severe adhesions from previous cervical surgery were primarily treated at the C7/T1 level. Furthermore, even if the incidence is small and no statistically significant difference is found, it may be desirable to perform the procedure in a safer location (larger epidural space) to avoid life-threatening complications. We summarized the most significant recent investigations of cervical interlaminar epidural steroid injection for the treatment of cervical radicular pain in Appendix A.

In two previous case reports of spinal cord injury after CIESI, patients were unable to report pain or irritation when the needle touched the spinal cord because of the effects of sedation [7,8]. Therefore, another recommendation is to avoid the routine use of sedatives during CIESI [6,7,8,12]. If venous sedation is considered essential, only the minimum dose required to achieve relief of anxiety should be administered to permit the patient to respond and report symptoms, which may indicate whether the needle is eroding the spinal cord. The patient in this study did not receive sedation during CIESI; however, this was a young patient who underwent CIESI for the first time. The patient reported severe burning and electric-like pain in the left arm during CIESI, which was recognized as pain that could occur during the procedure. Therefore, even for patients who do not receive intravenous sedation, it is necessary to check whether specific symptoms occur through communication with the patient, particularly for those undergoing the procedure for the first time.

Moreover, the contralateral oblique technique can be used to secure the epidural space. The needle should be inserted from the opposite side to secure the maximum space for it to enter the epidural space [6,12]. Additionally, predicting the intravascular location using aspirated blood is not considered reliable [18]. The use of live fluoroscopy to prevent intravascular penetration during interventions, such as CIESI, is considered essential [2,6,12]. However, physicians should recognize that fluoroscopy does not completely prevent intrathecal perforation or spinal infiltration during epidural steroid injections [7,8]. In addition, the use of particulate steroids should be avoided because of concerns regarding precipitation and aggregation, which can lead to vascular occlusion [12,19].

Changes in intracranial pressure due to inadvertent epidural puncture can result in SDH due to rupture of the dural connecting veins caused by brain movement [20]. In the patient in this case, the assessment by the imaging specialist was of a “a probable small amount of parafalcine SDH, left,” on the brain MRI performed 2 days after the procedure with cord injection. However, it was almost impossible to confirm the volume of SDH on MRI; therefore, SDH-related conservative treatment was recommended after consultations with neurosurgeons. The SDH volume was maintained without further treatment or exacerbation of symptoms. No findings suggestive of SDH-related sequelae were reported until the 6-month follow-up visit. However, when spinal cord injury during CIESI is suspected, brain MRI should be performed to differentiate the injury from SDH or other complications.

The mechanism of needle-induced nerve damage in CIESI can be explained as follows: needle stimulation induces local ischemia and nerve edema and deforms nerve fibers, resulting in local demyelination or axonal damage [21]. In addition, carriers used in some intrathecally injected steroids can be directly toxic to the central nervous system and cause injury [12]. For both the patient in this report and the patient in the case report by Hodges et al., C7/8 paresthesia persisted after needle insertion and steroid administration at C5/6 for up to 6 months of follow-up [7]. This may indicate that the patients’ neurological damage was not only a result of direct spinal cord penetration by the needle, but that it was also caused by a combination of adverse effects derived from exposure of the nerve tissues to the injected drugs (steroids and preservatives).

Intravenous and oral corticosteroids have strong anti-inflammatory effects and are often used to treat patients with nerve damage and inflammation, including nerve edema. A study revealed that dexamethasone promotes the regeneration of damaged sciatic nerves in a rat model [22]. Methylprednisolone sodium succinate was also reported to improve sustained neurological recovery in acute spinal cord injury in a phase 3 randomized trial [23]. Therefore, steroid pulse therapy can be used in patients with suspected nerve damage. Treatment is recommended with an initial intravascular bolus of 30 mg/kg for 15 min within 8 h of injury, followed by continuous infusion at 5.4 mg/kg/h for 24 h after 45 min [23]. Additionally, a 48 h extension in the time to maintenance therapy was found to result in further improvements in motor function recovery. This is particularly evident when the initial bolus can only be administered within 3–8 h of the injury [23]. In this case, approximately 25 h was required from the occurrence of the injury at another hospital to confirm nerve damage on the cervical spine MRI at our hospital. Steroids were administered 26 h after the procedure, which may have reduced the likelihood of improving the patient’s neurological deficits. Anticonvulsants and antidepressants are used to reduce neuropathic pain [24]. Among analgesics, tramadol has been reported to be more effective in reducing neuropathic pain than general nonsteroidal anti-inflammatory drugs or other opioids [24]. Therefore, after the injury, the patient in this case continued to receive steroids, anticonvulsants, antidepressants, and tramadol as maintenance drugs. She showed some improvement in nerve damage symptoms during the follow-up period.

According to another study, >85% of patients with PDPH achieved headache relief with conservative treatment within the first 24–48 h [25]. PDPH was suspected in our patient, and a blood patch was planned because of persistent PDPH after brain and cervical spine MRI. However, as PDPH was almost completely resolved after 48 h, a blood patch was not applied.

This study has several limitations. The patient in this report had a spinal cord injury after undergoing CIESI at another hospital and visited our hospital thereafter. Information regarding circumstances at the time of the procedure that caused the spinal cord injury was obtained from the patient and the physician or judged based on the procedure records; therefore, the exact conditions of the procedure were unknown. In addition, the patient was followed up for only 6 months. A longer follow-up period may be needed to determine whether a patient’s neurological deficits later resolve completely.

## 4. Conclusions

This report indicates that fluoroscopy does not guarantee the prevention of spinal cord penetration during epidural steroid injections. In addition, persistent neurological deficits may occur, particularly intrathecal perforation or drug administration during CIESI. Therefore, in accordance with the recommendations of the Multisociety Pain Workgroup, we strongly suggest performing CIESI at the C6/7 or C7/T1 level, where the epidural space is relatively large, rather than at the C5/6 level or higher.

## Figures and Tables

**Figure 1 medicina-58-01237-f001:**
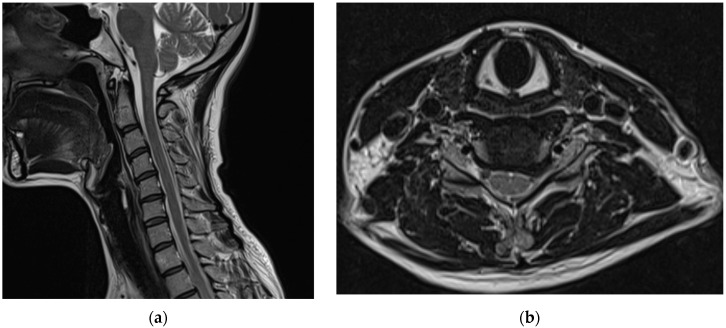
Cervical spine magnetic resonance images obtained before the procedure. A herniated intervertebral disc at the C5/6 level and left C6 nerve impingement due to a left foraminal extrusion are observable on the (**a**) T2-weighted sagittal image and (**b**) T2-weighted axial image of the C5/6 level.

**Figure 2 medicina-58-01237-f002:**
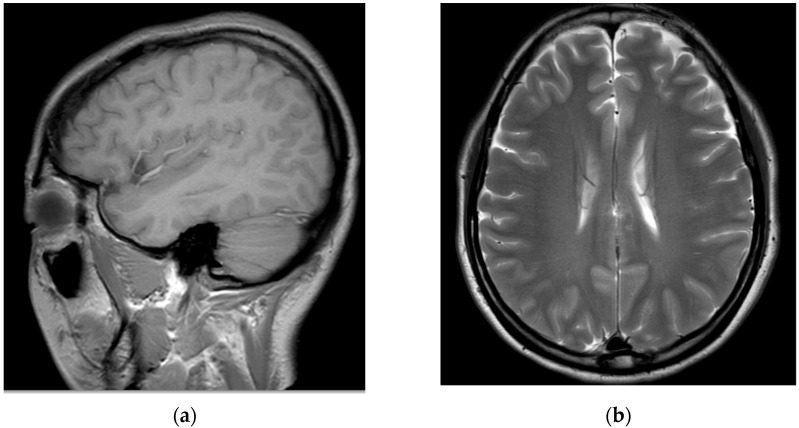
Brain magnetic resonance images obtained on the second day of the procedure: (**a**) T2-weighted sagittal image and (**b**) T2-weighted axial image. The assessment by the radiologist was of a “probable small amount of parafalcine subdural hematoma, left.” However, clear findings indicating a subdural hematoma are absent.

**Figure 3 medicina-58-01237-f003:**
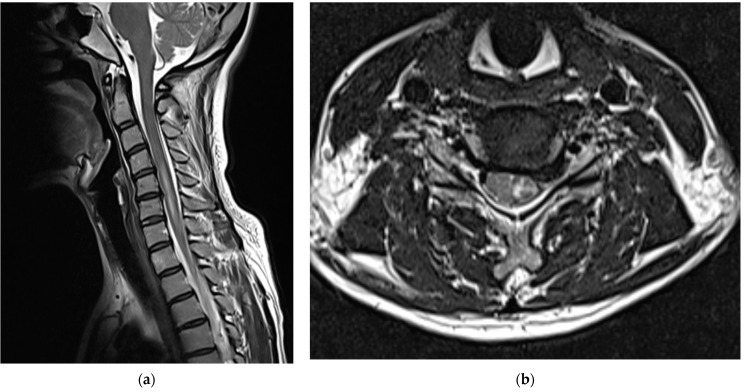
Cervical spine magnetic resonance images obtained on the second day of the procedure showed intramedullary T2 high signal intensity and cord swelling from C4/5 to C6/7 levels, which led to the diagnosis of a spinal cord injury: (**a**) T2-weighted sagittal images and (**b**) T2-weighted axial images of the C5/6 level.

**Figure 4 medicina-58-01237-f004:**
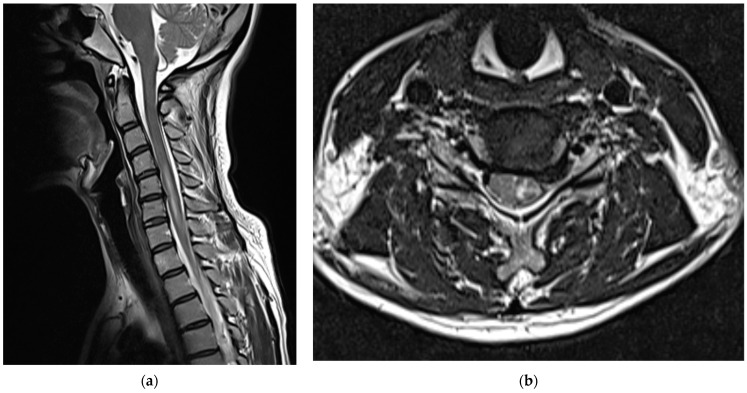
Cervical spine magnetic resonance images obtained 6 months after the procedure: (**a**) T2-weighted sagittal image and (**b**) T2-weighted axial image of the C5/6 level. The extent of the left central intramedullary T2 high signal intensity (elongated lesion in the spinal cord) from the C4/5 to C7 levels slightly reduced. The spinal cord swelling is also reduced.

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
