# Peer review of "Spinal Cord Injury and Postdural Puncture Headache following Cervical Interlaminar Epidural Steroid Injection: A Case Report"

_medicina, 2022, doi:10.3390/medicina58091237_

Round 1

Reviewer 1 Report

1. The topic of the m/s is relevant, describing a neurological complication after of the cervical interlaminar epidural steroid injection. In states clinical as well as imaging findings after the procedure explaining the neurological deficit. 2. The discussion is detailed providing possible explanation fort the complication as well as recommendations fot safer performance of the procedure, which is the main strenght of the m/s. As main weakness of the m/s I would state only superficial annotations of images without arrows as well as weak Legends to the Figures. Since authors are not radiologists nor is the Journal subspecialized for radiology, I would not demand from them to make major changes in this area. The m/s is well written in good English. 3. I found only minor issue to be corrected (same statement in two paragraphs). I.192-196 and I.218-222 are mostly repeating same statements. Suggest to combine or delete one of both paragraphs.

Author Response

MEDICINA-1851607

Spinal cord injury and postdural puncture headache following cervical interlaminar epidural steroid injection: A case report

Medicina

Response to reviewers’ comments

We appreciate the reviewers for their thoughtful comments and suggestions. We have revised the manuscript based on these recommendations, and we believe that these changes have improved the quality of our manuscript. We have also provided point-by-point responses and descriptions of these changes.

We thank the reviewers and the editor for their time.

Sincerely,

Chung Hun Lee

Department of Anesthesiology and Pain Medicine

Korea University Medical Center, Guro Hospital

Gurodong Road 148, Guro-Gu, Seoul 08308, Republic of Korea

Review Comments to the Author

Reviewer #1

  1. The topic of the m/s is relevant, describing a neurological complication after of the cervical interlaminar epidural steroid injection. In states clinical as well as imaging findings after the procedure explaining the neurological deficit:
  2. The discussion is detailed providing possible explanation fort the complication as well as recommendations fort safer performance of the procedure, which is the main strengh of the m/s. As main weakness of the m/s I would state only superficial annotations of images without arrows as well as weak Legends to the Figures. Since authors are not radiologists nor is the Journal subspecialized for radiology, I would not demand from them to make major changes in this area. The m/s is well written in good English.

-> Thank you for pointing this out. We have added the levels of the axial images in Figures 1, 3, and 4 to the figure legend. Thank you.

  • Figure 1. Cervical spine magnetic resonance images obtained before the procedure. A herniated intervertebral disc at the C5/6 level and left C6 nerve impingement due to a left foraminal extrusion are observed on the (a) T2-weighted sagittal image and (b) T2-weighted axial image of the C5/6 level.
  • Figure 3. Cervical spine magnetic resonance images obtained on the second day of the procedure shows intramedullary T2 high signal intensity and cord swelling from C4/5 to C6/7 levels, which led to the diagnosis of spinal cord injury: (a) T2-weighted sagittal images and (b) T2-weighted axial images of the C5/6 level.
  • Figure 4. Cervical spine magnetic resonance images obtained 6 months after the procedure: (a) T2-weighted sagittal image and (b) T2-weighted axial image of the C5/6 level. The extent of the left central intramedullary T2 high-signal intensity (elongated lesion in the spinal cord) from the C4/5 to C7 levels is slightly reduced. Spinal cord swelling is also reduced.

  1. I found only minor issue to be corrected (same statement in two paragraphs). I.192-196 and I.218-222 are mostly repeating same statements. Suggest to combine or delete one of both paragraphs.

-> Thank you for pointing this out. We avoided duplication by deleting page 6, lines 214-217 (underscore) on page 6, lines 192-196 and page 6, lines 214-217, which contain the same content, as observed by the reviewer. Thank you.

Reviewer 2 Report

This is a well written, interesting and very detailed case report.

Since the initial procedure was not performed at the author’s hospital, it is better to start with how the patient presented in a very bad state to the emergency department of the author’s hospital. Then, upon collecting the history, what happened earlier that led to the spinal cord injury can be described. This will make it clearer to the readers that the initial procedure was not done in the author’s hospital.

Upon reading Section 2, it is not very clear that the patient initially went to a different hospital for the CIESI procedure.

Are there any records of whether the CIESI was performed under fluoroscopy or ultrasound guidance? More details regarding this will be beneficial.

Figure 3 and 4 – Which cervical level does the axial image represent? This can be mentioned on the image.

Just a suggestion - In the discussion part, many studies are being referenced and discussed. It would be great for the readers if the authors can provide a table of all these studies that they are referring to. For Eg, see below

Author

Year

Type of Paper

Key messages

XXX et al.

XXX

Case Report

Hemiparesis and facial sensory loss after C5/6 interlaminar injection

XXX

Case Report

Patient symptomatically better; C7/8 paresthesia present for over 6 months

XXX

Original article

>85% patients achieve pain relief with conservative treatment

Author Response

MEDICINA-1851607

Spinal cord injury and postdural puncture headache following cervical interlaminar epidural steroid injection: A case report

Medicin

Response to reviewers’ comments

We appreciate the reviewers for their thoughtful comments and suggestions. We have revised the manuscript based on these recommendations, and we believe that these changes have improved the quality of our manuscript. We have also provided point-by-point responses and descriptions of these changes.

We thank the reviewers and the editor for their time.

Sincerely,

Chung Hun Lee

Department of Anesthesiology and Pain Medicine,

Korea University Medical Center, Guro Hospital,

Gurodong Road 148, Guro-Gu, Seoul 08308, Republic of Korea

Review Comments to the Author

Reviewer #2

This is a well written, interesting and very detailed case report.

- Thank you for your kind comment and your review.

Since the initial procedure was not performed at the author’s hospital, it is better to start with how the patient presented in a very bad state to the emergency department of the author’s hospital. Then, upon collecting the history, what happened earlier that led to the spinal cord injury can be described. This will make it clearer to the readers that the initial procedure was not done in the author’s hospital.

Upon reading Section 2, it is not very clear that the patient initially went to a different hospital for the CIESI procedure

- Thank you for pointing this out. We affirm that the initial procedure (CIESI) was not performed at the author's hospital, as the reviewer stated. However, we were concerned that if the authors, who are not native speakers of English, describe the patient’s previous medical history from the time of presentation at our hospital emergency room, it might further confuse readers. The contents of the case are described in chronological order; however, in line with the reviewer's suggestion, the contents of the text have been modified to improve the clarity of the report by adding that the patient's CIESI procedure, which was initially performed at another hospital. Thank you.

Are there any records of whether the CIESI was performed under fluoroscopy or ultrasound guidance? More details regarding this will be beneficial.

- Thank you for pointing this out. According to the medical records at the spinal hospital, the CIESI procedure was performed under fluoroscopic guidance. This detail has been added page 2, lines 79. Thank you.

Figure 3 and 4 – Which cervical level does the axial image represent? This can be mentioned on the image.

- Thank you for pointing this out. We have added the levels of the axial images in Figures 1, 3, and 4 to the figure legend. Thank you.

  • Figure 1. Cervical spine magnetic resonance images obtained before the procedure. A herniated intervertebral disc at the C5/6 level and left C6 nerve impingement due to a left foraminal extrusion are observed on the (a) T2-weighted sagittal image and (b) T2-weighted axial image of the C5/6 level.
  • Figure 3. Cervical spine magnetic resonance images obtained on the second day of the procedure shows intramedullary T2 high signal intensity and cord swelling from C4/5 to C6/7 levels, which led to the diagnosis of a spinal cord injury: (a) T2-weighted sagittal images and (b) T2-weighted axial images of the C5/6 level..
  • Figure 4. Cervical spine magnetic resonance images obtained 6 months after the procedure: (a) T2-weighted sagittal image and (b) T2-weighted axial image of the C5/6 level. The extent of the left central intramedullary T2 high-signal intensity (elongated lesion in the spinal cord) from the C4/5 to C7 levels is slightly reduced. Spinal cord swelling is also reduced.

Just a suggestion - In the discussion part, many studies are being referenced and discussed. It would be great for the readers if the authors can provide a table of all these studies that they are referring to. For Eg, see below

- Thank you for your suggestion. As suggested, we have provided an appendix of the important studies cited in the discussion section. Thank you.

Appendix A. Results of the most significant recent investigations on cervical interlaminar epidural steroid injection for the treatment of cervical radicular pain

Author

Year

Type of Paper (Design)

Level of CIESI

Key messages (Result)

Schneider et al. [6]

2018

Review article

Risks of CIESI are mitigated by using the contralateral oblique technique, limiting the use of sedation, injection at C6-7 or below, sterile technique, and following anticoagulation guidelines.

Hodges et al. [7]

1998

Case Report

C5/6

First case: C7/8 paresthesia present for over 6 months

Second case: severe refractory reflex sympathetic dystrophy of the left arm and paresthesia of the right thigh at 9 months after the procedure.

 Maddela et al. [8]

2014

Case Report

C5/6

Hemiparesis and facial sensory loss after C5/6 interlaminar injection

Restoration of facial sensation after 3 months; Motor function of the right leg continued to decline to Grade 4 by 3 months.

Rathmell et al. [12]

2015

Review article

The CIESI procedure should preferably be performed in C7-T1. Use sedatives minimally. A test dose of contrast medium is essential.

 Schultz et al. [15]

2022

Original article (Retrospective study)

C2/3

C3/4

C4/5

C5/6,

C6/7,

C7/T1

 Experience With 12,168 Procedures.

Complication rates did not increase with cervical injections cephalad to C7-T1.

Limitation: They did not mention the proportion of patients who had undergone cervical spine surgery at the C5/6, C6/7, and C7/T1 levels, and did not correct the results

 Manchikanti et al. [16]

2015

Original article (Retrospective study)

C5/6,

C6/7,

C7/T1

In 4396 patients, dural puncture rates were 1.8% for C5/6, 0.87% for C6/7, and 1.71% for C7/T1.

Limitation: They did not mention the proportion of patients who had undergone cervical spine surgery at the C5/6, C6/7, and C7/T1 levels, and did not correct the results

CIESI; Cervical interlaminar epidural steroid injection, C; Cervical

References

  1. Schneider, B.J.; Maybin, S.; Sturos, E. Safety and complications of cervical epidural steroid injections. Phys Med Rehabil Clin N Am. 2018, 29, 155–169.
  2. Hodges, S.D.; Castleberg, R.L.; Miller, T.; Ward, R.; Thornburg, C. Cervical epidural steroid injection with intrinsic spinal cord damage. Two case reports. Spine. 1998, 23, 2137–2142.
  3. Maddela, R.; Wahezi, S.E.; Sparr, S.; Brook, A. Hemiparesis and facial sensory loss following cervical epidural steroid injection. Pain Physician. 2014, 17, E761–E767.
  4. Rathmell, J.P.; Benzon, H.T.; Dreyfuss,P.; Huntoon, M.; Wallace, M.; Baker, R.; Riew, K.D.; Rosenquist, R.W.; Aprill, C.; Rost, N.S.; et al. Safeguards to prevent neurologic complications after epidural steroid injections: consensus opinions from a multidisciplinary working group and national organizations. Anesthesiology. 2015, 122, 974–984.
  5. Schultz, D.M.; Hagedorn, J.M.; Abd-Elsayed, A.; Stayner, S. Safety of interlaminar cervical epidural injections: experience with 12,168 procedures in a single pain clinic. Pain Physician. 2022, 25, 49–58.
  6. Manchikanti, L.; Malla,Y.; Cash, K.A.; Pampati, V. Do the gaps in the ligamentum flavum in the cervical spine translate into dural punctures? An analysis of 4,396 fluoroscopic interlaminar epidural injections. Pain Physician. 2015, 18, 259–266.